# Prognostic Genetic Biomarkers Based on Oncogenic Signaling Pathways for Outcome Prediction in Patients with Oral Cavity Squamous Cell Carcinoma

**DOI:** 10.3390/cancers13112709

**Published:** 2021-05-30

**Authors:** Wen-Lang Fan, Lan-Yan Yang, Jason Chia-Hsun Hsieh, Tsung-Chieh Lin, Mei-Yeh Jade Lu, Chun-Ta Liao

**Affiliations:** 1Genomic Medicine Core Laboratory, Linkou Chang Gung Memorial Hospital, Taoyuan 33382, Taiwan; alangfan@gmail.com (W.-L.F.); tclin1980@gmail.com (T.-C.L.); 2Clinical Trial Center, Biostatistics and Informatics Unit, Linkou Chang Gung Memorial Hospital, Taoyuan 33382, Taiwan; lyyang0111@gmail.com; 3Department of Internal Medicine, Division of Hematology-Oncology, New Taipei Municipal TuCheng Hospital, New Taipei City 236043, Taiwan; wisdom5000@gmail.com; 4Medical Oncology, Linkou Chang Gung Memorial Hospital, Taoyuan 33382, Taiwan; 5College of Medicine, Chang Gung University, Taoyuan 33382, Taiwan; 6Biodiversity Research Center, Academia Sinica, Taipei 11529, Taiwan; meiyehlu@gmail.com; 7Department of Otorhinolaryngology, Head and Neck Surgery, Linkou Chang Gung Memorial Hospital and Chang Gung University, Taoyuan 33382, Taiwan

**Keywords:** oral cavity squamous cell carcinoma, oncogenic signaling pathway, pathway instability, mutational signatures, disease-free survival, overall survival

## Abstract

**Simple Summary:**

A comprehensive analysis based on mutational signatures and oncogenic signaling pathways to identify a specific subgroup of patients that had a significantly negative impact on both disease-free and overall survival in oral cavity squamous cell carcinoma (OCSCC) from whole exome-sequencing data. This analysis has revealed a variety of biologically relevant candidate target genes. Thirty percent of 165 tumors had multiple targetable alterations in multiple pathways. This suggests the complex interplay and crosstalk of oncogenic signaling pathways play an important role on the outcome of patients with OCSCC, and the candidate genes and pathways identified may include prognostic genetic biomarkers or therapeutic targets for OCSCC.

**Abstract:**

Mutational profiling of patients’ tumors has suggested that the development of oral cavity squamous cell carcinoma (OCSCC) is driven by multiple genes in multiple pathways. This study aimed to examine the association between genomic alterations and clinical outcomes in patients with advanced stages OCSCC to facilitate prognostic stratification. We re-analyzed our previous whole-exome sequencing data from 165 long-term follow-ups of stages III and IV patients with OCSCC. Their frequent mutations were mapped to 10 oncogenic signaling pathways. Clinicopathological risk factors, relapse, and survival were analyzed to identify the genetic factors associated with advanced OCSCC. Frequent genetic alterations included point mutations in TP53, FAT1, NOTCH1, CASP8, CDKN2A, HRAS, PIK3CA, KMT2B (also known as MLL4), and LINC00273; amplified segments in CCND1, EGFR, CTTN, and FGFR1; and lost segments in CDKN2A, ADAM3A, and CFHR1/CFHR4. Comprehensive analysis of genetic alterations revealed that subgroups based on mutational signatures had a significant negative impact on disease-free survival (*p* = 0.0005) and overall survival (*p* = 0.0024). Several important signaling pathways were identified to be frequently genetically altered in our cohort. A specific subgroup of patients with alterations in NOTCH, RTK/RAS/MAPK, and TGF-beta pathways that had a significantly negative impact on disease-free survival (*p* = 0.0009). Thirty percent of samples had multiple targetable mutations in multiple pathways, indicating opportunities for novel therapy.

## 1. Introduction

Oral cavity squamous cell carcinoma (OCSCC) is the most common subtype of head and neck cancer worldwide [1]. It is thought to develop through a multistep carcinogenic process, including oncogene activation and loss of tumor suppressor genes. Locoregional recurrence, distant metastases, and the occurrence of second primary cancer are the leading causes of death in the post-treatment phase. OCSCC is closely related to tobacco smoking, alcohol drinking, and betel quid chewing [2,3,4]. However, the exact mechanisms of relapse and development of second primary cancer post-treatment remain to be elucidated. Recent advances in next-generation sequencing (NGS) of head and neck squamous cell carcinoma (HNSCC) samples of different etiologies have revealed that HNSCC development is mainly driven by loss-of-function alterations, including gene mutations in *TP53, CDKN2A, HRAS, PIK3CA, BRAF, CASP8*, and *NOTCH1* [5,6,7,8,9]. Another genomic analysis of OCSCC revealed five driver pathways (Notch, P53, cell cycle, Wnt, and mitogenic signaling) and defined novel molecular subtypes of OCSCC, characterized by frequent CASP8 mutations accompanied by copy-number alterations [8,10].

In this study, we explored the contribution of somatic genetic mutations to OCSCC at three analysis levels. First, we sequenced all exon-coding and exon-intron regions from samples of 165 patients with OCSCC through whole-exome sequencing (WES) to identify the frequent and significant mutations associated with OCSCC. Second, to further increase the analysis power, we mapped these mutations to well-known oncogene pathways in a previous study [11], which included 10 oncogenic signaling pathways curated from a total of 9125 cancers across 33 prevalent tumor categories, with the clinical factors associated with significantly mutated genes in patients with OCSCC and determined the significant association in each pathway. Third, to explore the influence of the OCSCC genetic variants on multiple pathways, disease-free survival (DFS) and overall survival (OS) were used as clinical endpoints to reveal correlations between genotype and clinical outcome. Collectively, our study systematically investigated the complex interplay among the multiple signaling pathways of OCSCC in the Taiwanese population.

## 2. Materials and Methods

### 2.1. Dataset

The original dataset included broad clinical data and exome sequences generated in our previous study [12]. In brief, we conducted a retrospective analysis of previously collected data. The study cohort consisted of 165 previously untreated consecutive patients with first primary OCSCC enrolled between February 2007 and July 2016. All patients were scheduled to undergo radical surgery with or without neck dissection (ND). The samples were obtained immediately after surgical resection from the operation theater, snap-frozen in liquid nitrogen, and stored at −80 °C. Restaging was performed in all patients according to the American Joint Committee on Cancer (AJCC) Staging Manual, eighth edition [13]. The clinical research protocol was reviewed and approved by the institutional review board of Chang Gung Memorial Hospital, Linkou, Taiwan (approval numbers: 99-4031B, 102-3136A3, and 102-0008A3). One author (Chun-Ta Liao) reviewed the clinical charts to collect patient demographics, pathological risk factors, and relapse events.

### 2.2. DNA Extraction and Whole-Exome Sequencing

Genomic DNA was extracted from fresh frozen tissues using the DNeasy Blood and Tissue Kit (Qiagen, Hilden, Germany) and sonicated with a hydrodynamic shearing system M220 Focused-ultrasonicator (Covaris, Woburn, MA, USA) to generate 200–280 bp fragments. A total of 24,156 genes and coding region sequences were captured using the Agilent exome enrichment kit (SureSelect Human All Exon v6, Agilent Technologies, Santa Clara, CA, USA) following the manufacturer’s recommendations. The exome-captured libraries were sequenced on an Illumina HiSeq 4000 system (Illumina, San Diego, CA, USA) at 150 bp paired-end reads.

### 2.3. Somatic Mutation Calling

Illumina sequencing raw data were quality trimmed and filtered using a fastp [14]. The remaining reads were aligned to the human genome assembly GRCh38 (hg38) using Burrows-Wheeler Aligner [15] using “BWA-MEM -t 32 -c 100 -M -R” and GATK [16] best practice. A panel of normal (PON) files was created using GATK version 4.1.2 [16] with aligned BAM files of matched non-tumor samples. MuTect2 was used to detect somatic mutations by comparing each OCSCC tumor with its matched normal sample and the PON file. The post-filtering step applied by the following criteria that was based on a visual inspection of read alignments by the Integrative Genomics Viewer [17]. First, mutations must covered by ≥8 aligned reads (mapping quality MAQ ≥ 20) in the tumor and matched normal samples, without variant reads in the matched normal samples. We then processed the different types of mutation using different settings. For single nucleotide variants (SNVs), mutations with a “FilterMutectCalls” filter flag “PASS” were selected. To improve accuracy in the mutation, the number of high-quality variant reads (mapping quality ≥ 30 and sequence base quality ≥ 20) were measured in the tumor and the number of variant reads in the matched normal sample. Only the SNVs supported by ≥2 high-quality variant reads in the tumor and no variant reads in the matched normal sample were selected. For small insertions and deletions (InDels), we only selected mutations with a “FilterMutectCalls” filter flag “PASS” supported by ≥8% variant reads in the tumor sample. The final PASS variants were annotated using VEP [18] and vcf2maf (https://github.com/mskcc/vcf2maf accessed on 1 September 2020).

### 2.4. Copy-Number Alteration Analysis

Somatic copy number variation were evaluated by using the CNVkit [19] pipeline. Because of the uncertainty in assessing the extent of copy number alteration in each sample, a log_2_ threshold of 0.3 was applied to detect for copy numbers gains or losses in the target regions. A heatmap of CNAs was generated using the heatmap command with the “-d” option to de-emphasize low-amplitude segments from the files of the CNVkit. We subsequently applied the GISTIC2.0 pipeline to detect significant amplification and deletion regions with somatic copy number alterations (CNAs) [20]. In this study, the cut-off q-value of GISTIC2.0 was set at 0.01.

### 2.5. Selection and Classification of Genes in Pathways

Mutated genes were assigned to oncogenic signaling pathways based on data from The Cancer Genome Atlas (TCGA) paper [11]. The “OncogenicPathways” module in maftools [21] was used to examine the enrichment of ten canonical oncogenic signaling pathways from the pathway analysis, including cell cycle, P53, RTK/RAS/MAPK, Hippo, Notch, PI-3-Kinase/Akt(PI3K), Myc, NRF2, Wnt and TGF-beta signaling pathways by the TCGA pan-cancer consortia. To calculate the fraction of genes/samples affected, we divided the number of genes/samples affected in each oncogenic signaling pathway by the number of genes/samples within that pathway. Variations that affect the active domains of a protein, such as missense, frameshift, non-frameshift, and spacing mutation, were used as inputs for signaling pathway analysis in the “OncogenicPathways” module.

### 2.6. Mutational Signature Analysis

Mutational signatures analysis was applied the non-negative matrix factorization (NMF) method [22] to extract signals in 165 OCSCC WESs. A permutation-based test was performed to apply mutation patters from the pan-cancer analysis of the whole-genome network (PCAWG) mutation signature analysis [23]. The classification of single-base substitutions (SBS) signatures were categorized into 96 classes of trinucleotide-based context (4 possible adjacent bases at 5′ × 6 possible base changes × 4 possible adjacent bases at 3′ is equal to 96). Doublet-base substitutions (DBS) signatures also were categorized into 78 distinct classes according to the identity of the doublet-bases alteration. InDels (ID) signatures based on a classification of InDels into 83 features according to whether they were small insertions and/or deletions (>100 bases) and the degree of homology with the surrounding sequences. The decomposed signatures were assigned to COSMIC signatures.

### 2.7. Significantly Mutated Genes

Significantly mutated genes (SMGs) were evaluated using the MutSig Covariate (MutSigCV) algorithm [24]. The MutSigCV algorithm detected SMGs with higher mutation occurrences than what was expected by chance. The covariate factors include the gene’s length, background mutation rate and base composition. This analysis was performed on all somatic mutations form 165 WES samples. The false discovery rate threshold of MutSigCV q-value was cut off at 0.1 (q < 0.1). Co-occurring or mutually exclusive sets of genes were detected using a pair-wise Fisher’s exact test in maftools [21].

### 2.8. Statistical Analysis

Statistical analysis was performed using free statistical software R (version 4.0.1). The “mafSurvival” module in maftools [21] was used to analyze overall survival and disease-free survival. The Kaplan–Meier method was used for survival analysis based on a grouping of mutation status in genes or pathways. The hazard ratio (HR) and odds ratio (OR) with 95% confidence intervals (CI) were used to evaluate the risk of survival outcomes and clinical characteristics. The *p*-value < 0.05 is considered statistically significant for all statistical tests.

## 3. Results

### 3.1. OCSCC Patient Samples

One hundred and sixty-five paired fresh-frozen malignant and adjacent non-malignant tissue specimens from treatment-naïve patients with resected OCSCC were analyzed for genetic alterations by using the WES approach. The clinicopathological details of 165 OCSCC cases in our cohort are described in detail in Appendix A. In brief, this cohort comprised 92.1% male, 82% alcohol drinker, 87% betel chewers, and 87% cigarette smokers, with a median age of 52 years (mean: 51 years, range 31–89 years). Among all the patients, 12% had p-Stage III and 88 % had p-Stage IV OCSCC tumors (Appendix A). The primary treatment modality in patients were surgical treatment followed by adjuvant chemoradiation or/and radiation therapy. The follow-up data of all patients were available, and the median survival time for our cohort was 41 months (mean: 44 months, range, 1–150 months).

### 3.2. Somatic Mutations and Mutational Signatures

To identify the genomic alterations in Taiwanese patients with OCSCC who had poor outcomes after surgical treatment, we conducted a WES of tumor and matched non-cancerous oral tissue samples from 165 patients with OCSCC (all of whom had long-term follow-up). A total of 165 tumors and matched normal samples were sequenced to average depths of 67.4-fold (range 101.9- to 44.5-fold; Appendix A, Appendix A) and 67.5-fold (range 102.8- to 42.7-fold; Appendix A, Appendix A), respectively. We mapped the sequence reads to the human reference genome (GRCh38) and identified a total of 17,662 somatic SNVs and 944 InDels after stringent filtering, with 2 to 407 mutations in sequencing targets per sample (Appendix A). All OCSCCs showed a low SNVs and InDels mutation burden, with an average of 1.48 mutations per Mb (range 0.03–4.71, Appendix A). Mutation signature analysis of 96 substitution patterns identified five signatures in the 165 OCSCC samples (Figure 1C, Appendix A). Five single base substitution (SBS) signatures closely matched known reference patterns in COSMIC signatures. (SBS1, SBS2, SBS5, SBS10b, and SBS13, nomenclature as in Alexandrov et al. [23]). Signature SBS1 was characterized by dominant C > T mutations, which are known to result from NCG trinucleotides and are probably a result of deamination of 5-methylcytosine. Its mutational load correlated linearly with age and may therefore be a cell division/mitotic clock. Signatures SBS2 and SBS13 are both characterized by C > G and C > T mutations at TCN, and are probably caused by the AID/APOBEC cytidine deaminase family [22,23]. The age-related clock-like signature SBS5 is a featureless base substitution and relatively featureless “flat” profiles of uncertain cause. Signature SBS10b is associated with polymerase epsilon (POLE) exonuclease domain deficiency, which is subjected to interactions with different POLE^exo^ mutations and microsatellite instability (MSI) factors. Additionally, a new doublet base substitutions and a small InDel (ID) mutational signature was found (Figure 1). These may relate to specific mutagenesis processes in OCSCCs, although the data required to confirm this are not currently available.

### 3.3. OCSCC Tumor Group (OTG) Subdivided with SBS Signatures

Agglomerative hierarchical clustering was conducted based on the proportions of five signatures of single base substitution (SBS1, SBS2, SBS5, SBS10b, and SBS13) in each sample (Figure 1A,B), which identified four OCSCC tumor groups (OTG), including two major groups of OCSCC tumors (OTG1 and OTG2, Figure 1A,B, Appendix A) from 165 samples. OTG1 and OTG2 were particularly enriched for SBS5 (*p* = 1.08 × 10^−4^, Appendix A) and SBS1 (*p* = 6.73 × 10^−4^, Appendix A), respectively. To explain the clinical significance of the two OTG groups, we examined the association between these mutations and clinical factors using both univariate and multivariate analyses of disease-free survival (DFS) and overall survival (OS) rates (Table 1, Appendix A). Patients in the OTG1 group typically contained more mutated genes, especially those associated with recurrent NOTCH1 and DNAH5 mutations (*p* = 1.08 × 10^−4^ and *p* = 6.73 × 10^−4^, respectively). In the univariate analysis, patients in the OTG1 group had a significantly negative impact on both DFS (*p* = 4.71 × 10^−4^, hazard ratio = 2.19) and OS (*p* = 2.38 × 10^−3^, hazard ratio = 2.2).

### 3.4. Recurrent Copy Number Events

Log-ratios of copy number were computed using CNVkit [19] with binary alignment maps (BAMs). Among the significant copy number alterations (CNAs) scored by GISTIC2.0, we identified 42 amplified segments, which harbored several known oncogenes such as EGFR (7p11.2), CCND1 (11q13.3), CTTN (11q13.3), and FGFR1 (8p11.22). We also identified 44 lost segments, which harbored tumor suppressors including CDKN2A (9p21.3), ADAM3A (8p11.22), and CFHR1/CFHR4 (1q31.3) (Appendix A, Figure 2, Appendix A). The two most frequent genes with copy-number changes, CCND1 (44 samples, 26.7%) and/or EGFR (34 samples, 20.6%), were found in 100 out of 165 (60.6%) OCSCC tumors, among which 45 (27.3%) carried more than a segment change. The amplification of the well-known oncogenes CCND1 and EGFR was associated with a metastatic lymph node, as shown by the OS and DFS in a previous oral squamous cell carcinoma (OSCC) study in Taiwan [25,26]. In contrast, there were no significant differences in OS (*p* = 0.198 and *p* = 0.669) or DFS (*p* = 0.256 and *p* = 0.226) in our cohort. CNAs are more commonly associated with loss of function in tumor suppressor genes, most prominently CDKN2A, which is frequently mutated in HNSCC and OSCC [5,6,7,8,9]. In our OCSCC cohort, patients with CDKN2A gene mutations and/or deletions had significantly worse OS (*p* = 0.0012), and those with RICTOR gene deletion had significantly worse DFS (*p* = 0.0068) (Appendix A).

### 3.5. Significantly Mutated Genes and Pathway Analysis

Significantly mutated genes in the 165 OCSCC samples were identified using MutSigCV [24]. Mutations in TP53 (108 samples, 65.5%), FAT1 (48 samples, 29.1%), NOTCH1 (39 samples, 23.6%), CASP8 (23 samples, 13.9%), CDKN2A (22 samples, 13.3%), HRAS (12 samples, 7.3%), and PIK3CA (12 samples, 7.3%) were found to be significant (q < 0.01) in the 165 OCSCC cases (Figure 2, Table 1, Appendix A). In addition, KMT2B (also known as MLL4) and LINC00273 were frequently mutated in more than 6.6% of the 165 samples, which may also be driver genes in OCSCCs. A total of 153 tumors (92.73%) had somatic mutations in at least one of the 25 significantly mutated genes listed (Figure 2, Appendix A), including a significant copy number alterations, which can be subdivided into ten oncogenic pathways (Appendix A), including P53 (109 samples, 66.1%), Ras/Raf/MAPK (86 samples, 52.1%), Hippo (79 samples, 47.9%), cell cycle (72 samples, 43.6%), Notch (72 samples, 43.6%), PI3K (33 samples, 20.0%), Wnt (28 samples, 17.0%), MYC(8 samples, 4.8%), TGF-beta (6 samples, 3.6%), and NRF2 (3 samples, 1.8%) signaling pathways.

### 3.6. Co-Occurrence and Mutual Exclusivity

The co-occurrence and mutual exclusion of gene mutations may influence prognosis; therefore, we explored the co-occurrence and mutual exclusivity of mutations. To examine significantly co-occurring and mutually exclusive alterations by gene or pathway, we used the “somaticInteractions” function of maftools [21] to examine the mutations found in 165 OCSCC tumors (Appendix A). Somatic mutations in TP53, FAT1, and NOTCH1 frequently co-occurred in our cohort. Among the 35 significant genes, coexisting mutations were identified in 80.61% (133/165) of patients, while only one mutation was found in 93.94% of patients with mutations (155/165). The top 50 high-frequency gene pairs of co-occurring and mutually exclusive mutations are shown in Appendix A. We noted weak mutual exclusivity between TP53 and NOTCH1 mutations (*p* = 0.1830), in contrast to the co-occurrence of somatic mutations in NOTCH1 and CDKN2A (*p* = 0.0017), SYNE1 and CDKN2A (*p* = 0.0038), CASP8 and DNAH5 (*p* = 0.0054), and NOTCH1 and CASP8 (*p* = 0.0066). We also found that 61.36% of all mutations in CCND1 occurred in combination with a second oncogenic mutation (usually in EGFR), and that oncogenic mutations in the HRAS gene often occur in combination with a mutation in the RICTOR gene (3/12, 25%). In addition, mutations in CDKN2A, CASP8, HRAS, and DNAH5 co-occurred with NOTCH1 mutations (Appendix A). Patients with OCSCC typically have multiple functional alterations that affect more than a single pathway, which is similar to a previous study which examined 33 other cancers [11]. The co-occurring pathways detected here by a set of pathway gene mutations exhibited a statistically significant co-occurrence in the samples. Upon mapping these mutations to the affected oncogenic signaling pathways, we found that the most frequently co-occurring pathways were between P53 and RTK/RAS/MAPK.

### 3.7. Survival Analysis for Mutations, Pathways, and Clinical Factors

To clarify the clinical outcomes of these driver-mutated genes and pathways, we examined the association of these genetic alterations with OTGs, pathways, clinical factors, and prognosis of the 165 OCSCC cases using both univariate and multivariate analyses of OS and DFS (Figure 3A–D and Figure 4, and Appendix A). In the univariate analysis, none of the clinical factors (age, betel nut chewing, cigarette smoking, and alcohol consumption) showed any significant differences in OS and DFS. There was no statistically significant difference in DFS or OS between patients with no mutations and those with one, two, or more significant mutations in their tumors (Appendix A, Appendix A). Only the negative effects of single genes on OS were observed in OCSCC patients harboring mutations in CDKN2A (n = 36, *p* = 0.0012, Figure 3B). Notably, we found co-occurring pathways between P53 and NOTCH signaling pathways (51 samples, *p* = 0.0081), RTK/RAS/MAPK and NOTCH (43 samples, *p* = 0.0018, Figure 3D), RTK/RAS/MAPK and cell cycle (42 samples, *p* = 0.0008), which had a significantly negative impact on OS. Hippo and NOTCH (43 samples, *p* = 0.0075), NOTCH and PI3K (18 samples, *p* = 0.0076), NOTCH and WNT (14 samples, *p* = 0.0025, Figure 3C), cell cycle and PI3K (18 samples, *p* = 0.0051) had a significantly negative impact on DFS (Table 2). Among the concurrent alterations in the three signaling pathways, we identified a specific subgroup of patients with alterations in NOTCH, RTK/RAS/MAPK, and TGF-beta pathways (Figure 4) that had a significantly negative impact on both OS (*p* = 0.0120, Figure 4A) and DFS (*p* = 0.0009, Figure 4B).

## 4. Discussion

In the current study, we described the somatic mutation landscape in a unique set of advanced-stage patients with pathological stage IV OCSCC (145 samples, 87.9%) using whole-exome sequencing. The genomic landscape of OCSCC is dominated by the loss of tumor suppressor genes, most frequently TP53, FAT1, and CDKN2A, each of which is lost in the majority of tumors. CASP8, which was the fourth most frequently inactivated gene, is a surprising new candidate for tumor suppressor. There was a strong correlation among CASP8, HRAS, NOTCH1, and DNAH5 mutations in our OCSCC cohort (Appendix A). CASP8 and HRAS mutant tumors were generally a subset of the NOTCH1 mutant tumors, suggesting that HRAS and CASP8 mutations are permissive for NOTCH1 mutations. The aspartate-specific cysteine protease (caspase)-8 is an aspartate-specific cysteine protease (ASCPs) that plays an important role in the initiation and regulation of death receptor-mediated apoptosis [27]. Oncogenic HRAS sensitized fibroblasts to TRAIL-induced apoptosis. In addition, CASP8 cleavage and CASP8, which were also found frequently (11.2%) in head and neck squamous cell carcinomas [10], are associated with poor overall survival in univariate and multivariate analyses [28].

Histone-lysine N-methyltransferase 2D (KMT2D; also known as MLL2) and histone-lysine N-methyltransferase 2 B (KMT2B; also known as MLL4), both encoding histone-lysine N-methyltransferases, were frequently mutated in HNSCC and OSCC [5,6,7,8,9]. Mago homolog B (MagohB) transcriptional silencing in the absence of KMT2B is characterized by an increase in DNA methylation over the MagohB CpG island promoter and a loss of trimethylation of Histone H3 at Lysine 4 (H3K4me3) [29]. KMT2D is a major mammalian histone H3K4 mono-methyltransferase along with di-methyltransferase. Both KMT2B and KMT2D play important roles in controlling bulk histone 3 lysine 4 methylation (H3K4me) during oocyte growth and preimplantation development. Additionally, a long intergenic non-protein coding RNA 273 (LINC00273) was also frequently mutated in our cohort (11 samples, 6.7%). LINC00273, as an inducer of metastasis, which significantly inhibits epithelial to metastasis and mesenchymal transition, promotes cancer metastasis. The G-quadruplex promoter of LINC00273 may play a pivotal role in the prognosis and serve as a new biomarker and therapeutic target for OCSCCs tumor invasion [30,31].

Our new findings suggest that aberrations in the transforming growth factor beta (TGF-β) pathway may promote carcinogenesis in OCSCC. A total of 3.6% of patients (n = 6) with mutated genes in the TGF-β signaling pathway had a significantly negative impact on OS and DFS. Comparing OTG1 with OTG2, we found that FAT atypical cadherin 1 (FAT1) mutations were strongly associated with betel nut chewing in our cohort. FAT1 regulates extracellular matrix architecture and cell adhesion while acting as a tumor suppressor in oral cancers in a context-dependent manner. Despite the clinicopathologic implication of FAT1 in many malignant tumors, its function in OCSCCs or OSCCs has not been yet clarified. Herein, we documented that the driver oncogene FAT1 may play an important role in patients with OCSCC who have the habit of chewing betel nut.

A total of 143 samples (86.7%) had one or more actionable genes, representing alternative treatment opportunities for surgically treated patients with adverse survival outcomes. The genome of OCSCC contains many alterations, including frequent oncogenic drivers and candidate therapeutic targets. With more knowledge derived from studies using approaches such as pathway instability analyses, it may be possible to make individualized treatment decisions that can improve the survival of patients with OCSCC.

## 5. Conclusions

We identified several prognostic genetic biomarkers that were enriched in certain subtypes and others that affected both disease-free survival and overall survival. This study identified a variety of biological relevant targets, some of which vary by OCSCC cancer subtype. Here we report a large-scale whole-exome sequencing association study of OCSCC and clinical outcomes among 165 individuals, and aids in resolving the inherent heterogeneity of OCSCC. Since eight samples (4.8%) did not harbor any of the cancer-related mutated genes or copy number alterations identified, the data showed that the remaining gaps in our knowledge of associations between mutated genes and OCSCCs, and other genomic alterations can drive OCSCC development. Of the 25 frequently and significantly mutated genes identified, 11 were actionable according to the CIViC database [32] (https://civicdb.org accessed on 18 November 2020). Thirty percent of tumors had multiple targetable alterations in multiple pathways, indicating opportunities for novel therapy. This suggests a complex interplay between different oncogenic signaling pathways, indicative of pathway crosstalk and multi-pathway instability in OCSCC. Herein, we got more knowledge derived from this study using the approach of oncogenic pathway analysis, but our research was conducted in a betel quid chewing endemic area, which may limit the generalizability of our findings.

## Figures and Tables

**Figure 1 cancers-13-02709-f001:**
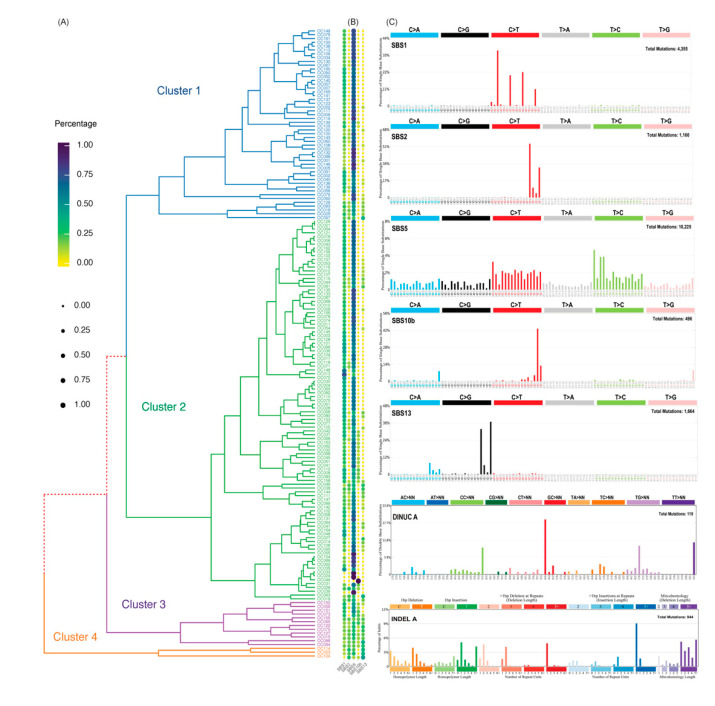
Mutational signatures present in 165 oral cavity squamous cell carcinoma (OCSCC) samples. (**A**) Tumor classification (OCSCC tumor group, OTG1–4) based on the contributions of mutational signatures in each tumor is represented by a colored line in the dendrogram. (**B**) The percentage attributed to each signature is shown in dotmatrix. (**C**) An example each of single base substitution (SBS), doublet base substitution (DBS; also labelled as DINUC), and small insertion or deletion (ID) signatures showing the categories into which mutations are divided. These figures of mutation patters are shown with the color scheme, ordering, and categories into which the mutations are divided. The complement of signatures discovered in 165 OCSCC. Known COSMIC signatures are marked according to their nomenclature, while novel signatures are marked with “DINUC A” and “INDEL A”.

**Figure 2 cancers-13-02709-f002:**
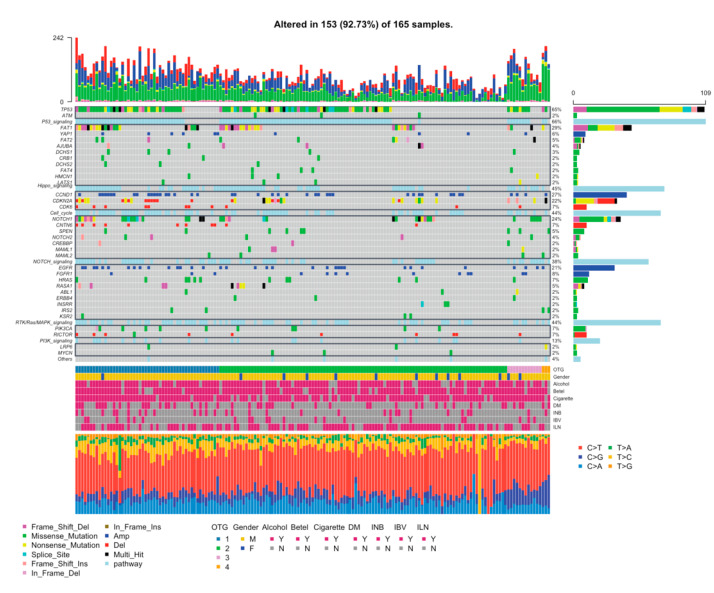
Significantly and frequently mutated genes in 165 oral cavity squamous cell carcinoma (OCSCC) samples detected by exome sequencing. Each column represents one OCSCC sample and each row represents the overall survival status or mutated genes of each sample. A total of 165 OCSCCs are described here, and oncogenic pathways to which each gene belongs are shown on the left.

**Figure 3 cancers-13-02709-f003:**
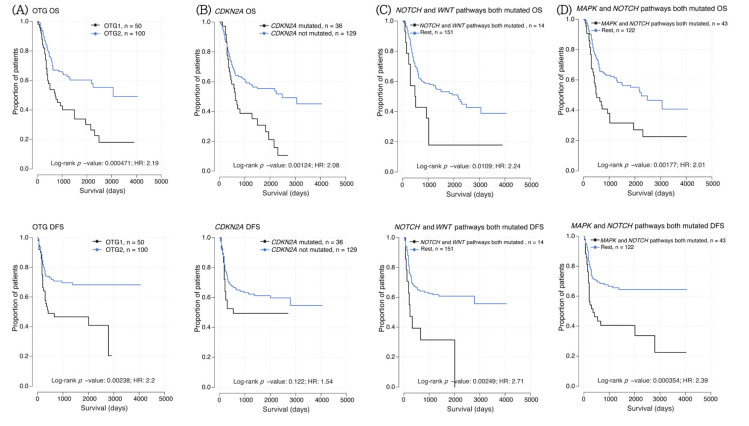
Impact of specific mutations on disease-free survival (DFS) and overall survival (OS) and in oral cavity squamous cell carcinoma. Kaplan–Meier survival curves for OS and DFS based on patients with mutations in (**A**) OCSCC tumor group 1 and 2, (**B**) CDKN2A, (**C**) NOTCH and WNT signaling pathways, (**D**)NOTCH and TGF-beta signaling pathways, respectively.

**Figure 4 cancers-13-02709-f004:**
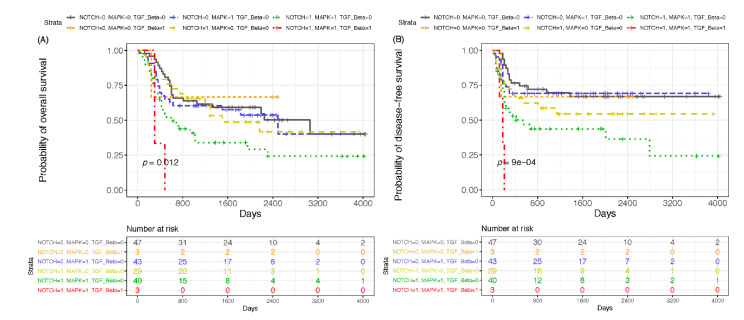
Overall survival (OS) and disease-free survival (DFS) according to gene alterations in three pathways. Kaplan–Meier survival curves for (**A**) OS and (**B**) DFS based on patients with mutations in NOTCH, RTK/RAS/MAPK, and TGF-beta signaling pathways. Plots were generated using the “survival” package version 3.2-3 of R software (r-project.org version 4.0.1).

**Table 1 cancers-13-02709-t001:** Frequently mutated genes in oral cavity squamous cell carcinoma. Del indicates a deletion present in the gene being transcribed. Amp indicates an amplification in the gene.

Gene	Mutation Number and Frequency in OTG1 (*n* = 50)	Mutation Number and Frequency in OTG2 (*n* = 100)	Odd Ratio	Fisher’s *p*-Value	Mutation Number and Frequency in 165 OCSCCs	MutSigCV *p*-Value	MutSigCV *q*-Value
TP53	38 (76%)	60 (60%)	2.11	0.038	108 (65%)	0	0
FAT1	21 (42%)	23 (23%)	2.42	0.014	48 (29%)	0	0
NOTCH1	19 (38%)	15 (15%)	3.47	0.002	39 (24%)	1.44 × 10^−15^	1.70 × 10^−12^
CASP8	6 (12%)	15 (15%)	0.77	0.409	23 (14%)	0	0
CDKN2A	5 (10%)	13 (13%)	0.74	0.404	22 (13%)	0	0
PIK3CA	6 (12%)	5 (5%)	2.59	0.113	12 (7%)	7.72 × 10^−5^	6.33 × 10^−2^
HRAS	6 (12%)	5 (5%)	2.59	0.113	12 (7%)	0	0
MUC5B	5 (10%)	6 (6%)	1.74	0.283	11 (7%)	2.31 × 10^−1^	1
LINC00273	6 (12%)	4 (4%)	3.27	0.070	11 (7%)	NA	NA
DNAH5	7 (14%)	2 (2%)	7.98	0.007	10 (6%)	2.90 × 10^−2^	1
MUC4	6 (12%)	3 (3%)	4.41	0.038	10 (6%)	1.44 × 10^−3^	4.16 × 10^−1^
FAT2	3 (6%)	5 (5%)	1.21	0.535	9 (5%)	5.04 × 10^−3^	7.37 × 10^−1^
KMT2B	4 (8%)	2 (2%)	4.26	0.096	9 (5%)	NA	NA
PLEC	3 (6%)	5 (5%)	1.21	0.535	9 (5%)	1.58 × 10^−1^	1
RASA1	4 (8%)	5 (5%)	1.65	0.347	9 (5%)	0	0
SPEN	2 (4%)	4 (4%)	1.00	0.683	9 (5%)	1.73 × 10^−2^	1
SYNE1	4 (8%)	2 (2%)	4.26	0.096	9 (5%)	5.17 × 10^−1^	1
CCND1 Del	19 (38%)	21 (21%)	2.31	0.023	44 (27%)	-	-
EGFR Del	15 (30%)	15 (15%)	2.43	0.027	34 (21%)	-	-
FGFR1 Del	3 (6%)	8 (8%)	0.73	0.469	13 (8%)	-	-
YAP1 Del	6 (12%)	3 3%)	4.41	0.038	10 (6%)	-	-
CDKN2A Amp	10 (29%)	4 (4%)	6.00	0.003	14 (8%)	-	-
CDK6 Amp	8 (16%)	2 (2%)	9.33	0.002	11 (7%)	-	-
RICTOR Amp	4 (8%)	5 (5%)	1.65	0.347	10 (6%)	-	-
CNTN6 Amp	9 (18%)	2 (2%)	10.76	0.001	11 (7%)	-	-

**Table 2 cancers-13-02709-t002:** Clinical impact of specific pathway mutations on overall survival and disease-free survival in oral cavity squamous cell carcinoma. DFS, disease-free survival; OS, overall survival; HR, hazard ratio. Bold indicated statistically significant findings (*p* < 0.05).

Number of Mutated Pathways	Variable Pathways	n/N (DFS)	DFS HR	DFS *p*-Value	n/N (OS)	OS HR	OS *p*-Value
1	P53 signal	109/165	1.15	0.58	109/165	0.89	0.61
1	RTK/RAS/MAPK signal	79/165	1.28	0.32	88/165	1.51	0.06
1	Hippo signal	86/165	1.50	0.10	76/165	0.91	0.64
1	NOTCH signal	72/165	0.46	1.25 × 10^−3^	72/165	0.02	0.61
1	Cell cycle	72/165	1.40	0.17	72/165	1.64	0.02
1	PI3K	33/165	0.59	0.05	33/165	0.69	0.14
1	WNT	28/165	0.82	0.52	28/165	0.87	0.63
2	P53 and RTK/RAS/MAPK	58/165	1.54	0.05	58/165	1.54	0.05
2	P53 and Hippo	56/165	1.25	0.37	56/165	1.12	0.61
2	P53 and NOTCH	51/165	1.79	0.02	51/165	1.77	8.05 × 10^−3^
2	P53 and Cell cycle	53/165	1.45	0.13	53/165	1.66	0.02
2	P53 and PI3K	24/165	1.79	0.05	24/165	1.75	0.04
2	P53 and WNT	23/165	1.05	0.87	23/165	1.05	0.87
2	RTK/RAS/MAPK and Hippo	48/165	1.36	0.25	48/165	1.10	0.68
2	RTK/RAS/MAPK and NOTCH	43/165	2.39	3.54 × 10^−4^	43/165	2.01	1.77 x 10^−3^
2	RTK/RAS/MAPK and Cell cycle	42/165	1.61	0.07	42/165	2.11	7.78 × 10^−4^
2	RTK/RAS/MAPK and PI3K	20/165	1.49	0.23	20/165	1.41	0.28
2	RTK/RAS/MAPK and WNT	21/165	1.31	0.43	21/165	1.35	0.32
2	Hippo and NOTCH	43/165	1.95	7.54 × 10^−3^	43/165	1.23	0.37
2	Hippo and Cell cycle	43/165	1.52	0.12	43/165	1.23	0.38
2	Hippo and PI3K	17/165	1.76	0.10	17/165	1.12	0.75
2	Hippo and WNT	17/165	1.24	0.56	17/165	1.03	0.95
2	NOTCH and Cell cycle	38/165	1.78	0.03	38/127	1.67	0.03
2	NOTCH and PI3K	18/165	2.29	7.63 × 10^−3^	18/165	1.89	0.03
2	NOTCH and WNT	14/165	2.71	2.49 × 10^−3^	14/165	2.24	0.01
2	Cell cycle and PI3K	18/165	2.38	5.13 × 10^−3^	18/165	1.99	0.02
2	Cell cycle and WNT	14/165	1.16	0.69	14/165	1.28	0.48
2	PI3K and WNT	6/165	0.80	0.75	6/165	0.64	0.52
3	P53, RTK/RAS/MAPK, and NOTCH	33/165	2.09	5.71 × 10^−3^	33/165	2.22	5.87 × 10^−4^
3	P53, RTK/RAS/MAPK, and Cell cycle	31/165	1.60	0.10	31/165	2.22	6.49 × 10^−4^
3	P53, RTK/RAS/MAPK, and WNT	17/165	1.26	0.52	17/165	1.39	0.31
3	P53, NOTCH, and Cell cycle	30/165	1.70	0.06	30/165	1.68	0.03
3	P53, NOTCH, and WNT	11/165	2.59	0.08	11/165	2.21	0.02
3	P53, Cell cycle, and WNT	12/165	1.12	0.77	12/165	1.27	0.52
3	RTK/RAS/MAPK, NOTCH, and Cell cycle	26/165	2.08	9.57 × 10^−3^	26/165	2.17	1.93 × 10^−3^
3	RTK/RAS/MAPK, NOTCH, and WNT	12/165	2.37	0.02	12/165	1.95	0.05
3	RTK/RAS/MAPK, Cell cycle, and WNT	11/165	1.37	0.45	11/165	1.69	0.16
3	NOTCH, Cell cycle, and WNT	12/165	1.12	0.77	12/165	1.27	0.52

## Data Availability

Data of this study will be available upon request.

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
