# Peer review of "Prognostic Genetic Biomarkers Based on Oncogenic Signaling Pathways for Outcome Prediction in Patients with Oral Cavity Squamous Cell Carcinoma"

_cancers, 2021, doi:10.3390/cancers13112709_

Round 1
Reviewer 1 Report
Overall the methodology of the work is appropriate to the objectives of the research and the findings are of interest for scientific community.
Additionally, the quality of the patient data set is very good and accomplish for the purpose of the work.
The methodology is accurate a and I appreciate that it was really well described in the entire manuscript.
The main concern to have in consideration before publication regards to the concept of the novelty that the authors have confer to the work. I would considered change somehow the title of the manuscripts: “Novel Prognostic Genetic Biomarkers Based on Oncogenic Signaling Pathways for Outcome Prediction in Patients with Oral Cavity Squamous Cell Carcinoma” as all the biomarkers that the authors have found out in the analysis have been already extensively described in previous work (some of them also cited by the authors).
In any case, I would like to congratulate the authors for the great work.
Reviewer 2 Report
Well written paper with probably one of the largest series among similar publication. The only limitation on the interpretation of the data is because of the high incidence of betel nut chewing in the cohort (Taiwan), the data may not be the same as other part of the world.
